# Consumer Awareness, Perceptions and Avoidance of Ultra-Processed Foods: A Study of UK Adults in 2024

**DOI:** 10.3390/foods13152317

**Published:** 2024-07-23

**Authors:** Eric Robinson, Jenna R. Cummings, Thomas Gough, Andrew Jones, Rebecca Evans

**Affiliations:** 1Institute of Population Health, University of Liverpool, Liverpool 69 7ZA, UK; jenna.cummings@liverpool.ac.uk (J.R.C.);; 2School of Psychology, Liverpool John Moores University, Liverpool L3 5AH, UK

**Keywords:** ultra-processed food, UPF, consumers, attitudes, perceptions

## Abstract

Background: Ultra-processed food (UPF) is currently not included in most countries’ dietary guidance. However, there may be growing public interest and consumer avoidance of UPF due to media reporting of studies on the negative health outcomes associated with UPFs. Methods: We surveyed 2386 UK adults (M age = 45 years, 50% female) during February–April 2024. Participants completed measures on awareness of the term UPF, whether the UPF status of foods affects their dietary decision-making, and confidence in identifying UPFs. Participants categorised a list of 10 foods (5 UPFs and 5 non-UPFs) as UPF vs. not, before rating whether information about studies linking UPF to worse health impacts on their negative affect and acts as a deterrent to consuming UPFs. Results: Most participants (73%) were aware of the term UPF and 58% reported that their food choices are determined by whether they believe a food is ultra-processed or not. Participants with the highest income and education levels were most likely to report both being aware of, and, avoiding consuming UPFs. Most participants could not accurately categorise whether foods were UPFs. Some sociodemographic groups (e.g., higher education levels) were more likely to accurately categorise UPFs but were also more likely to incorrectly believe that non-UPFs were UPFs. Participants tended to report that UPF-health risk information increases negative affect and acts as a deterrent to consuming UPFs. Conclusions: In this study, a large number of UK adults reported avoiding consuming UPFs. This was particularly pronounced among those with the highest education and income levels.

## 1. Introduction

The term ‘ultra-processed food’ (UPF) was first introduced in a scientific context in 2009 as part of the NOVA classification system [1]. The NOVA classification system characterises foods based on their level of industrial processing (i.e., whole food fractioning, chemical modification, assembly, additions for palatability, and packaging), with four categories ranging from unprocessed/minimally processed foods to UPF. The UPF category is characterised by food products which ‘are formulations of ingredients, mostly of exclusive industrial use, that result from a series of industrial processes [2]. According to [2], common examples of UPFs include confectionery, sodas, packaged breads and reconstituted meat products (e.g., hot dogs). It has also been suggested by [2] that a practical way to identify UPFs is to examine whether listed ingredients contain substances rarely used in home kitchens (e.g., hydrogenated oils) and/or include additives to enhance food palatability or appearance (e.g., sweeteners).

Although the NOVA classification system and utility of the term UPF has received some criticism (e.g., [3]), consumption of UPFs has been linked to a wide range of negative health outcomes in epidemiological studies [4,5]. However, there is scientific uncertainty concerning whether a food’s degree of processing causally impacts on health because most of the research to date has been observational in nature [6] and observational findings can be prone to confounding [7]. Similarly, it remains unclear whether UPFs are associated with worse health outcomes in observational studies due to the macronutrient profile of UPFs or for other reasons [8]. Irrespective of this uncertainty, there have been calls for public health policies to address UPFs and reduce their consumption [9].

Although it is common for national dietary guidance to recommend limiting consumption of foods high in fat, salt and/or sugars (which many UPFs are), few countries (with the exception of a small number of predominantly South American countries, such as Brazil) explicitly include terminology on food processing or UPFs in national dietary guidance [10].

Limited research has examined consumer perceptions of UPFs. In a 2016 study of Uruguayan adults, the majority of participants (92%) reported being aware of UPF [11]. A 2019 study of young adults in Argentina and Ecuador concluded that participants had relatively high levels of awareness of UPFs, but did not accurately identify foods as UPFs (or not) with a high level of consistency [12]. In a 2022 study of Brazilian adults, 82% of participants reported being aware of the term UPF, but ability to correctly identify products as UPFs was variable and dependent on product type [13]. A 2022 study asked participants about awareness of the NOVA classification. Among sampled Italian and Dutch participants, the majority were unaware of the NOVA classification system, whereas Brazilian participants were [14]. Studies to date have (i) been largely limited to South American countries, and/or (ii) examined awareness of the concept UPF and/or ability to correctly identify UPFs. Therefore, there is a lack of wider international evidence on consumer perceptions of UPFs and limited evidence on consumer responses to UPF.

In the UK, UPFs do not currently feature in national government dietary guidance or public health policy. However, there has been media coverage of research linking UPF consumption to negative health outcomes [15]. Similarly, the UK government have recently commissioned an evidence review on UPFs and health, as well as convening a parliamentary committee on the topic [6].

Given apparent growing public interest in UPFs, the aim of the present study was to examine UK consumers’ awareness and perceptions of them, including whether consumers report being influenced by the concept of UPF when making dietary choices. We also examined confidence and accuracy in identifying UPFs. Finally, given increasing media attention reporting on UPF negative health associations, a further objective was to examine the perceived impact of information about studies linking UPFs to worse health outcomes on negative affect (e.g., distress) and the extent to which this information acts as a deterrent to consuming UPFs.

## 2. Method

### 2.1. Sampling

During February–April 2024, participants were recruited as part of a larger online experiment examining perceived effectiveness and impact on hypothetical food choices of ‘high in salt’ warning labels on packaged food and restaurant menus. Participants were recruited and recruitment was stratified to be balanced and broadly representative of the UK population, resulting in strata based on participant gender (49:51 male vs. female), age (36:64 18–39 years vs. 40 years and over) and highest education level achieved (52:48 NQF level 4 or above vs. NQF level 3 or below: equivalent to university educated vs. not). Participants were recruited from a widely used online participant panel (Prolific) and compensated for their time. To be eligible for inclusion, participants had to be UK residents, aged 18 years or above and a fluent speaker of English. Based on the aims of the larger online experiment, participants who purchased supermarket sandwiches/savoury snacks or ate restaurant food at least monthly were eligible for inclusion. Participants who were pregnant, breast-feeding or reported dietary restrictions/intolerances (e.g., gluten-free, vegan, or vegetarian) were ineligible.

In the larger study, participants completed self-reported demographic questionnaires (see Table 1) before being randomised to complete hypothetical food choice tasks in the presence vs. absence of ‘high in salt’ warning labels differing in design (4 red/black octagon/triangle warning label conditions, 1 no warning label condition). Self-reported demographics included sex, ethnicity, age, weight and height in preferred units of reporting (to calculate BMI), highest education level achieved and monthly household income (including reporting of household composition to enable calculation of equivalised household income). Participants also completed other self-report measures on perceived effectiveness and acceptability of warning labels. The study included two attention checks (e.g., how many times have you visited the planet Mars?) and a consistency of responding check (education level reported at the beginning and end of the study) to identify unreliable responders and reduce responding bias. Participants completed the UPF-related measures described below at the end of the study. For full information (including design and measures included) on the larger online experiment that the present data were collected at the end of, see https://osf.io/vyh2x/ (accessed on 3 July 2024).

### 2.2. Measures

To measure awareness of UPF, as in previous studies [13] participants were asked ‘Have you ever heard of ultra-processed foods?’ and ‘Do you know what ultra-processed foods are?’ (response options: No, Yes, Unsure).

To measure the perceived influence of UPF on dietary choice, participants were asked ‘Do you think about whether a food is ultra-processed when deciding whether or not to eat it’? (response options: No, Yes—I try to avoid eating ultra-processed foods, Yes—I never eat any ultra-processed foods, Yes—I try to eat ultra-processed foods, Yes—I only ever eat ultra-processed foods). See Table 2.

Participants were then asked if they felt confident in identifying whether a food is an UPF, before on the next page being provided with a list of 10 names of foods and categorising each as UPF vs. not. Foods were chosen from NOVA guidance [2], resulting in five UPFs (baby formula, burgers, breakfast cereals, ice cream, packaged bread) and five non-UPFs from NOVA groups 1–3 (salted nuts, fruit in syrup, smoked meat, maple syrup, pasteurised yoghurt). Foods were selected on the basis of their name not including explicit information directly pertaining to processing or lack of (e.g., processed meat). See Table 3 for items and their NOVA classification.

Next, participants were shown the text: ‘Studies have found that consuming ultra-processed foods is associated with an increased risk of health conditions like cancer, diabetes, heart disease, obesity and early death’, based on typical UPF news coverage in the UK [15]. To examine self-perceived effects of information on negative affect, participants rated four negative affect items (how distressed, scared, afraid, anxious they felt in response to the text) from the Positive and Negative Affect Schedule (PANAS) [16], using a 5-point response scale (Very slightly to not at all, A little, Moderately, Quite a bit, Extremely). Using a validated measure of the extent to which public health information is perceived as being likely to change health behaviour [17], participants rated the extent to which the text discouraged them from wanting to consume UPFs, made them concerned about the health effects of consuming UPFs and that consuming ultra-processed foods seems unpleasant, using the same 5-point response scale.

### 2.3. Primary Analyses

Logistic regression was used to examine sociodemographic predictors of participants having heard of the term UPF (yes vs. no/unsure) and if their food choices are determined by whether a food is ultra-processed or not (yes vs. no/unsure). Linear regression was used to examine sociodemographic predictors of the number of UPFs and non-UPFs correctly categorised (0–10 total score), perceived impact of UPF-health information on negative affect (four items averaged, α = 0.96) and the extent to which UPF-health information acts as a deterrent to consuming UPFs (three items averaged, α = 0.94). Sociodemographic predictors were age (18–39 yrs, 40 yrs and over), sex, highest education level (university degree level vs. not), ethnicity (white vs. other), equivalised household income (quintiles) and BMI (<18.5, 18–24.9, 25–29.9, ≥30 kg/m^2^). For analysis purposes, age and education were each categorised into two groups to reflect the study sampling strata. Ethnicity was categorised into white vs. not due to small sample sizes in all non-white ethnicity categories. Alpha was set a 0.01 to account for multiple testing.

## 3. Results

A total of 2548 participants were recruited. After removal of 72 participants for failing an attention or consistency check, 85 for implausible BMI values (<10 or >60 kg/m^2^) and 5 for missing equivalised income data, the final analytic N = 2386.

Sample characteristics are reported in Table 1.

### 3.1. UPF Awareness and Consideration

As indicated in Table 2, the majority of participants reported having heard of UPF (73%), knowing what UPF is (57%) and thinking about whether a food is UPF when deciding whether to eat it (58%). From the total sample, 54% of participants reported avoiding eating UPFs, 3% reported never eating any UPFs, 1% reported trying to eat UPFs or only ever eating UPFs.

Participants with the highest education level, highest income quintile (vs. lowest quintile) and older adults (≥40 yrs) were both significantly more likely to report being aware of the term UPF and report thinking about whether a food is UPF when deciding what to eat (Table 4).

The only other significant associations for these outcomes were that white participants were more likely to report being aware of the term UPF and participants with obesity (vs. normal weight BMI) were less likely to report thinking about whether a food is UPF when deciding whether to eat it. Sociodemographic patterning for ‘knowing what UPFs are’ and confidence in identifying UPFs was similar to results for being aware of the term UPF (see Appendix A).

### 3.2. UPF Categorisation

A minority of participants (39%) reported feeling confident in identifying whether a food was UPF or not (Table 2). On average (mean), participants correctly categorised 2.7/5 UPFs and 3.5/5 non-UPFs. Only 13% of participants accurately categorised all UPFs. See Table 3 for accuracy of categorisation for individual UPF (ranging from 35−66%) and non-UPF items (ranging from 51% to 82%), in addition to percentages of participants accurately categorising all vs. no UPFs and non-UPFs. Having the highest education, household income and being white were associated with a greater total number of foods correctly categorised as UPF vs. not. However, when examining the accuracy of UPFs and non-UPFs categorisation separately, being female, having higher education levels and being of an older age were associated with both better UPF categorisation and worse non-UPF categorisation, highlighting a tendency among these participants to classify more foods (both correctly and incorrectly) as UPF (see Appendix A).

### 3.3. Perceived Impacts of UPF-Health Risk Information

Mean scores for the extent to which information about the health consequences of UPFs was perceived as impacting on negative affect (α = 0.96) and deters consuming UPF (α = 0.94) were close to the centre of each scale, denoting average response options of ‘a little’ to ‘moderately’ (Table 2). Females and non-white participants reported higher negative affect in response to the information. For perceptions of acting as a deterrent to consume UPF, males, older participants (≥40 yrs), white participants (vs. non-white), participants with obesity and lower education levels reported being less affected by the information (Table 4).

## 4. Discussion

In a large sample of UK adults, 73% reported being aware of the term UPF and 58% reported that their food choices are determined by whether they believe a food is ultra-processed or not. Of these, the majority reported actively avoiding consuming UPFs. However, only a minority of participants felt confident in identifying UPFs and when asked to categorise 10 foods based on being UPF vs. not, on average participants failed to consistently correctly categorise UPFs (based on NOVA classification). Participants with the highest education levels and household income were more likely to report being aware of UPFs and avoiding consuming them (ORs = 1.67–2.38). This is the first UK study to examine consumer perceptions of UPFs and the first study we are aware of to examine the extent to which consumers report avoiding UPFs. The relatively high levels of consumer awareness and self-reported avoidance of UPFs observed in this study are striking given the lack of national dietary guidance or public health policy on UPF in the UK and current scientific uncertainty over the causal impact that food processing has on health [6].

Inaccuracy in identifying UPFs is consistent with a study of French food and nutritional specialists [18]. In this study, although some UPFs were more consistently identified, overall there was a low level of accuracy [18]. In the present study, some sociodemographic groups (e.g., higher education levels) were more likely to accurately categorise UPFs as such. However, the same groups were also more likely to incorrectly categorise non-UPFs as UPFs, highlighting a tendency among these groups to classify more foods (both correctly and incorrectly) as UPF. Collectively these findings suggest that although there appears to be significant consumer interest in UPFs, levels of understanding of which foods are classed as UPF may be limited.

It is unclear why participants tended to incorrectly categorise some non-UPFs as UPF and UPFs as non-UPF. Participants may perceive many common staple foods (such as packaged bread) as relatively healthy due to their ubiquity and therefore do not consider them to be in the same category of processing as many foods typically thought of as ‘junk food’ (e.g., burgers). Additionally, consumers may be unaware of the degree of processing in many common foods. Research specifically designed to understand the features of foods which determine whether they are perceived as being UPF will be required to understand this.

News reports in the UK frequently report on the health risks associated with consuming UPFs [15] and we asked participants to what extent this information made them experience negative affect and acted as a deterrent to consuming UPFs. Data were consistent with the proposition that information linking UPF with worse health may be making consumers experience negative emotions towards UPFs and avoiding consuming them. Participants with obesity (as opposed to normal weight) were less likely to report avoiding UPFs and believed less strongly that information about the health risks associated with consuming UPFs makes eating them more unappealing and worrying. These results are consistent with the observation that obesity tends to be associated with health being a relatively less important motive for food choice [19].

In addition to the discussed sociodemographic patterning of awareness of the term UPF, categorization accuracy and reported avoidance of UPFs, there were also sociodemographic differences in the reported impact UPF-health information has on negative affect. Both females and non-white ethnic groups reported an increased negative affect compared to males and white participants. These groups (as well as older adults) were also more likely to report that UPF-health information acts as a deterrent to consuming UPFs. These findings suggest that public communication about UPFs may be having differential effects on the population and causing some distress among population subsections.

Collectively, the present findings suggest that even though there is currently no UK public health guidance to avoid consuming UPFs [6], large numbers of people in the UK may now be actively avoiding consuming UPFs out of concern for potential impacts on health. It will therefore be valuable to better understand how concerns over UPFs may be affecting the nutritional quality of dietary patterns and if there are any negative or positive consequences of recent public interest in and concern over UPFs. Likewise, further research understanding why UPFs are now being widely avoided would be informative. One theoretical position on food avoidance is that humans feel an inherent disgust towards and practise avoidance of foods they believe to be ‘contaminated’ and made ‘unnatural’ [20,21], so describing foods as UPF may evoke such tendencies.

### 4.1. Strengths and Limitations

The measures included were brief as they were taken as part of a larger study. Therefore, future work would benefit from collecting more detailed measures, including whether consumers provide accurate definitions of UPF and are able to accurately identify a wider range of UPFs and non-UPFs. For example, the list of non-UPFs participants categorised consisted of more processed foods (NOVA group 3) than NOVA groups 1 and 2, and the latter groups may be easier to identify as non-UPF. Similarly, the UPFs categorised were selected on the basis of their description not explicitly providing information about processing and not a random selection of UPFs and non-UPFs, which may have affected categorisation accuracy. The present study did not collect detailed dietary data or the specific UPFs that consumers now report avoiding (e.g., specific types of UPFs, or all UPFs non-discriminately). Future research examining consumer perceptions of UPFs in higher and lower habitual consumers of UPF would be informative. The use of validated measurement tools to examine consumer perceptions of UPFs will also be a priority of future research [22]. Perceived influence of information about the health risks associated with UPF consumption was self-reported and therefore we cannot conclude if this type of information has a causal impact on consumer behaviour. As is the case in all survey research, results may be biased by reporting biases, such as socially desirable responding. As the present study was conducted online (as opposed to with a researcher present), we presume socially desirable responding should be minimal. Confirmatory research measuring objective behaviour would now be valuable. Our sample was selected on the basis of eligibility criteria (e.g., eating at restaurants at least monthly) and excluded participants with dietary restrictions. As is the case in the UK, our sample was predominantly white. Overall, our sample also tended to be representative of the UK in terms of sex, age and BMI. However, our sample had a relatively higher level of education than the UK average and we were unable to examine if results differed between non-white ethnic groups, both of which are limitations of the present research. Further research will be required to determine if findings are generalisable to other population groups (e.g., ethnic minority groups). These limitations aside, the present research benefited from sampling a large and socio-demographically diverse panel of UK adults. Although the present research is preliminary, we have identified a striking overall level and sociodemographic patterning of consumer awareness and self-reported avoidance of UPFs among the general population. Future research is therefore warranted to better understand consumer awareness, avoidance and perceptions of UPFs, as well as the potential impact that growing public interest in UPF may have on dietary behaviour and health-related outcomes.

### 4.2. Conclusions

In this study, a large number of UK adults reported avoiding consuming UPFs. This was particularly pronounced among those with the highest education and income levels.

## Figures and Tables

**Table 1 foods-13-02317-t001:** Sample Characteristics.

	N (%) or M (SD)
	*N = 2386*
Sex (Female)	1202 (50%)
Ethnicity (White)	2085 (87%)
Age (M years, SD)	45.2 (13.5)
Age (40 years or older)	1539 (65%)
BMI (M, SD)	27.3 (6.1)
Underweight BMI (<18.5)	54 (2%)
Normal weight BMI (18.5–24.9)	938 (39%)
Overweight BMI (25–29.9)	791 (33%)
Obesity BMI (≥30)	603 (25%)
Education level (lower) ^a^	1219 (51%)
Equivalised household income (GBP M, SD)	£26,274 (17,382)

^a^ Education level (lower) denotes less than university degree education (NQF level 3 or below).

**Table 2 foods-13-02317-t002:** Participants’ UPF questionnaire responses.

*UPF Questionnaire Items*	*Yes*	*No*	*Unsure*
Have you ever heard of UPFs?	1737 (73%)	390 (16%)	259 (11%)
Do you know what UPFs are?	1356 (57%)	529 (22%)	501 (21%)
Do you think about whether a food is UPF when deciding whether to eat it?	1376 (58%)	1010 (42%)	-
Would you feel confident in identifying whether a food is UPF or not?	928 (39%)	742 (31%)	716 (30%)
* **Scoring of UPF measures** *	Mean (SD)		
Number of UPFs (/5) correctly identified	2.7 (1.5)		
Number of non-UPFs (/5) correctly identified	3.5 (1.3)		
Total number of UPFs and non-UPFs correctly identified (/10)	6.2 (1.6)		
Perceived influence on negative affect of UPF-health information ^a^	2.4 (1.2)		
Perceived deterrent of UPF-health information ^b^	3.2 (1.2)		

^a^ Four negative affect items (distressed, anxious, afraid, scared) on a 5-point Likert scale (1 = very slightly or not all, 5 = extremely) averaged. ^b^ Three perceived deterrent items (discouraged from wanting UPFs, concerned about health effects of UPFs, consuming UPFs seems unpleasant) on a 5-point Likert scale (1 = very slightly or not all, 5 = extremely) averaged.

**Table 3 foods-13-02317-t003:** Participant categorisation of individual UPFs and non-UPFs.

Food Items [NOVA Group in Brackets]	*Percent Correct ^a^*
*UPF items*	
Baby formula [group 4]	827 (35%)
Breakfast cereals [group 4]	1348 (57%)
Burgers [group 4]	1584 (66%)
Ice cream [group 4]	1209 (51%)
Packaged bread [group 4]	1378 (58%)
*Non-UPF items*	
Fruit in syrup [group 3]	1513 (63%)
Maple syrup [group 2]	1965 (82%)
Pasteurised yoghurt [group 1]	1896 (80%)
Salted nuts [group 3]	1884 (79%)
Smoked meat [group 3]	1212 (51%)

^a^ Percent correct is proportion of participants correctly categorising a food as UPF (or not). NOVA group 4 = ultra-processed, group 3 = processed, group 2 = culinary ingredients, group 1 = unprocessed or minimally processed.

**Table 4 foods-13-02317-t004:** Sociodemographic predictors of UPF questionnaire items and scores.

	Ever Heard of UPFs (Yes)	UPF Influence Food Choice (Yes)	UPF and Non-UPF Identification (/10)	UPF-Health Info Influence on Negative Affect (/5)	UPF-Health Info Acts as a Deterrent to Consume UPF (/5)
	Nagelkerke R^2^ = 0.089	Nagelkerke R^2^ = 0.069	Adjusted R^2^ = 0.021	Adjusted R^2^ = 0.027	Adjusted R^2^ = 0.031
	Odds Ratio	Odds Ratio	B	B	B
Sex (female)	0.90, *p* = 0.28	1.18, *p* = 0.06	0.02, *p* = 0.74	0.28, *p* < 0.001	0.20, *p* < 0.001
Ethnicity (white)	1.97, *p* < 0.001	0.98, *p* = 0.89	0.43, *p* < 0.001	−0.41, *p* < 0.001	−0.33, *p* < 0.001
Age (≥40 yrs)	1.63, *p* < 0.001	1.51, *p* < 0.001	−0.06, *p* = 0.41	−0.11, *p* = 0.03	0.14, *p* = 0.006
Underweight BMI ^a^	0.99, *p* = 0.99	0.76, *p* = 0.34	0.13, *p* = 0.56	−0.01, *p* = 0.95	−0.21, *p* = 0.21
Overweight BMI ^a^	0.88, *p* = 0.24	0.91, *p* = 0.36	−0.09, *p* = 0.22	0.06, *p* = 0.26	−0.02, *p* = 0.78
Obesity BMI ^a^	0.97, *p* = 0.83	0.69, *p* < 0.001	0.09, *p* = 0.25	−0.03, *p* = 0.65	−0.28, *p* < 0.001
Education (higher) ^b^	2.38, *p* < 0.001	1.87, *p* < 0.001	0.32, *p* < 0.001	0.05, *p* = 0.31	0.13, *p* = 0.01
Income quintile 2 ^c^	1.03, *p* = 0.85	1.20, *p* = 0.17	0.05, *p* = 0.64	0.03 *p* = 0.69	0.01, *p* = 0.89
Income quintile 3 ^c^	1.24, *p* = 0.15	1.22, *p* = 0.14	0.05, *p* = 0.60	0.06, *p* = 0.45	0.06, *p* = 0.46
Income quintile 4 ^c^	1.23, *p* = 0.16	1.29, *p* = 0.06	0.07, *p* = 0.50	0.04, *p* = 0.60	0.04, *p* = 0.58
Income quintile 5 ^c^	1.67, *p* = 0.001	1.70, *p* < 0.001	0.29, *p* = 0.005	0.04, *p* = 0.57	0.13, *p* = 0.10

^a^ Comparison category is normal weight BMI. ^b^ Comparison category is less than university education (NQF level 3 or below). ^c^ Comparison category is income quintile 1.

## Data Availability

The data presented in this study are available on request from the corresponding author. The datasets presented in this article are not readily available because the data are drawn from a larger study for which data analysis and publication are not complete.

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
