# Peer review of "Consumer Awareness, Perceptions and Avoidance of Ultra-Processed Foods: A Study of UK Adults in 2024"

_foods, 2024, doi:10.3390/foods13152317_

Round 1

Reviewer 1 Report (Previous Reviewer 4)

Comments and Suggestions for Authors

authors addressed well main remarks. No major concerns remain. 

Reviewer 2 Report (Previous Reviewer 1)

Comments and Suggestions for Authors I am happy that the authors have addressed my comments. I think that the paper can be accepted.

This manuscript is a resubmission of an earlier submission. The following is a list of the peer review reports and author responses from that submission.

Round 1

Reviewer 1 Report

Comments and Suggestions for Authors

Introduction – This is an interesting and timely topic to investigate. The introduction sets up the key issues which need to be addressed. It could perhaps include some discussion of what ultraprocessed means and the NOVA scale.

Method – The method of the study is described in considerable detail. It could perhaps be explained why most of the demographics are all collapsed into two categories. How were they measured? It would have been interesting to examine differences within the variables. In the method it does not mention collecting information about the BMI but this appears in the results. Similarly income emerges in the results and is not mentioned in the method.

Results – The results are presented clearly.

Discussion – It states that levels of awareness are limited  despite 73% being aware of UPF. It refers to a French study and doesn’t provide the reference. The discussion of the results is a bit haphazard. It could be a little more systematic in its approach. It does not go much beyond repeating the results which is a bit limiting. Why are people categorising non-UPFs as UPF? Why are there the variations in numbers correct  for the various UPF and non-UPF foods?

Author Response

We thank the reviewers for their constructive comments. Below we have outlined how we have accommodated each comment and where appropriate, have produced reference to where in the revised manuscript changes can be found.

Reviewer 1

1] Introduction – This is an interesting and timely topic to investigate. The introduction sets up the key issues which need to be addressed. It could perhaps include some discussion of what ultraprocessed means and the NOVA scale.

RESPONSE: We thank the reviewer for their positive comments.

2] Method – The method of the study is described in considerable detail. It could perhaps be explained why most of the demographics are all collapsed into two categories. How were they measured? It would have been interesting to examine differences within the variables. In the method it does not mention collecting information about the BMI but this appears in the results. Similarly income emerges in the results and is not mentioned in the method.

RESPONSE: We have now specified why demographics are collapsed into two categories for analysis purposes. See page 3.

We now also provide more detailed information on how variables were measured, including information on BMI and income, as suggested. See page 2.

3] Results – The results are presented clearly.

RESPONSE: We thank the reviewer for their positive comments.

4] Discussion – It states that levels of awareness are limited  despite 73% being aware of UPF. It refers to a French study and doesn’t provide the reference. The discussion of the results is a bit haphazard. It could be a little more systematic in its approach. It does not go much beyond repeating the results which is a bit limiting. Why are people categorising non-UPFs as UPF? Why are there the variations in numbers correct  for the various UPF and non-UPF foods?

RESPONSE: We have now revised the Discussion to make it clearer that levels of awareness refer to awareness of the term ‘UPF’, as opposed to levels of understanding of which foods are vs. are not classed as UPFs. We also now provide a reference immediately after referring to the French study (Braesco et al., 2022).

We are somewhat constrained by the word count for the article type of this submission. However, we have now provided some suggestions as to why there may be variations in accuracy of categorising UPFs and why it was common for some non-UPFs to be incorrectly classified as UPF. See Discussion, page 7. We have also added to the discussion to ensure that sociodemographic differences for all studied outcomes are now covered in the Discussion. See page 8.  

Reviewer 2 Report

Comments and Suggestions for Authors

Dear authors,

I have read and reread very carefully the article I received for review. The proposed research topic is topical and of great interest to researchers and science in general. I noticed an interesting aspect, at least by comparison with many articles that I have reviewed in various journals in the WOS category. It is about the fact that in the quantitative research carried out on a sample of 2386 people, you did not issue hypotheses, but wanted to discover what, how and why it happens this way and not otherwise. It is a new trend in research and I think it is welcome.

From my point of view, the article has a good scientific foundation, it is written in a high academic style, and the results are presented accurately. The research methodology is adequate.

Although it is an implicit one, I RECOMMEND to specify the PURPOSE of this research.

Author Response

We thank the reviewers for their constructive comments. Below we have outlined how we have accommodated each comment and where appropriate, have produced reference to where in the revised manuscript changes can be found.

Reviewer 2

1] I have read and reread very carefully the article I received for review. The proposed research topic is topical and of great interest to researchers and science in general. I noticed an interesting aspect, at least by comparison with many articles that I have reviewed in various journals in the WOS category. It is about the fact that in the quantitative research carried out on a sample of 2386 people, you did not issue hypotheses, but wanted to discover what, how and why it happens this way and not otherwise. It is a new trend in research and I think it is welcome.

RESPONSE: We thank the reviewer for their positive comments.

2] From my point of view, the article has a good scientific foundation, it is written in a high academic style, and the results are presented accurately. The research methodology is adequate.

RESPONSE: We thank the reviewer for their positive comments.

3]Although it is an implicit one, I RECOMMEND to specify the PURPOSE of this research.

RESPONSE: We have now outlined at the end of the Introduction what the primary aims and purpose of the present research was. See page 2.

Reviewer 3 Report

Comments and Suggestions for Authors

1.The sampling design is biased one which means that it only focused on certain group. For example, Ethnicity (White) (87%) that you can find in the table 1. In this case, ethnicity is not supposed to be discussed since you do not have other ethnic subgroup to compare with.

2.The same biased situation in the data discussion section is found. The authors had skipped several data, ex, the normal weight subgroup, no discussion on gender. The authors also should discuss the comparisons among the sociodemographic characteristics while they did not.

3.I failed to see the conclusion they made is based on enough scientific evidence.

Author Response

We thank the reviewers for their constructive comments. Below we have outlined how we have accommodated each comment and where appropriate, have produced reference to where in the revised manuscript changes can be found.

Reviewer 3

1] 1.The sampling design is biased one which means that it only focused on certain group. For example, Ethnicity (White) (87%) that you can find in the table 1. In this case, ethnicity is not supposed to be discussed since you do not have other ethnic subgroup to compare with.

RESPONSE: We now outline in the discussion that our results are based on a predominantly white sample and that further research will be required to determine generalisability. See page 8.

2] The same biased situation in the data discussion section is found. The authors had skipped several data, ex, the normal weight subgroup, no discussion on gender. The authors also should discuss the comparisons among the sociodemographic characteristics while they did not.

RESPONSE: We have now ensured that sociodemographic differences for all studied outcomes are now covered in the Discussion. See page 8. This includes an extended section on gender and ethnicity differences. The only sub-group difference identified which relates to the normal weight (vs. the obesity group) is discussed on page 8 also.

3] I failed to see the conclusion they made is based on enough scientific evidence

RESPONSE: We have made a number of revisions, including outlining limitations to the present research. We have also edited the conclusions section of our abstract to make it clear that our conclusions relate to the sampled participants in the present study. See Abstract, page 1.

Reviewer 4 Report

Comments and Suggestions for Authors

The work covers an hot topic with respect to public health and the environment. Even if the environmental dimension remains absent, the paper can be improved while focusing on public health. The objectives of the study are clear but the work would certainly attract attention from a broader audience by summarizing the main goal and specific objectives at the end of the introduction section. 

The work may benefit from a few more changes.

It is clear that artificialized foods (notably class 4 of Nova classification) are unhealthy and unsustainable as shown by mounting scientific studies. 

True foods are dense in nutrients and bioactive compounds and its composition is complex and undetermined in opposition to such industrial foods, known as UPF. 

However, the definition of UPF has been contested, especially by engineers for whom “processing” is synonym of “unitary operations”. Other authors highlight the confusion with formulation. The authors are thus suggested to present these conflicting views (suggested ex. Levine AS, Ubbink J. Ultra-processed foods: Processing versus formulation. Obes Sci Pract. 2023 Jan 26;9(4):435-439. doi: 10.1002/osp4.657. 

Knorr D, Watzke H. Food Processing at a Crossroad. Front Nutr. 2019 Jun 25;6:85. doi: 10.3389/fnut.2019.00085.)

Please avoid less clear information and update the concept/definition supported by recent literature. Therefore, at this level, authors may consider deleting the examples in line 35 “Common examples of UPFs include confectionary, ice cream, breakfast cereals, packaged bread and reconstituted meat products (e.g., hot dogs).” Instead, “junk food” and “sodas” are more clearer examples to be presented at this level, if authors believe it to be useful. Further clarification on class IV foods should be included at some level in the manuscript, because ice cream can be made of macerated fresh fruit, sacharose and often egg white and some corn flakes and oatmeal are processed in some way that are not included in class 4;. Lines 36-40 can also bring confusion as recently industries bring ultraprocessed ingredients to kitchens (ex. refined flour containing so-called “flour improvements” for home bread-making without the need for yeast fermentation). It can be also helpful for the reader if the authors also elaborate on processed foods, as those of class I, II, on nutritional security, convenience of use, extended shelflife vs. the poor nutritional features and negative impacts in public health and the environment of UPF. 

Methods: Please clearly state inclusion and exclusion criteria as well as the number of initial recruited participants and those who competed the study, and the initial wording can thus be summarised. 

Please include a scheme or table outlining the experimental design. Also include a table (different from that of supplementary material) clearly listing the different foods presented to participants, the corresponding short description, and the Nova class they belong to. Results: the discussion in the section UPF categorization can be enriched noticing the difficulties of participants in clearly understand the UPF concept and hence failing identifying class IV foods. Please elaborate and further explain results on table 3. Please also expand section 3.3. relating with the information from introduction (including newly inserted one). Please make the necessary adjustments on section 4.1 for coherence reasons. concluding remarks should be added; pls add a short conclusion section summarizing main findings, the relevance of the study and prospects.

Author Response

We thank the reviewers for their constructive comments. Below we have outlined how we have accommodated each comment and where appropriate, have produced reference to where in the revised manuscript changes can be found.

Reviewer 4

The work covers an hot topic with respect to public health and the environment. Even if the environmental dimension remains absent, the paper can be improved while focusing on public health.

RESPONSE: We thank the reviewer for their positive comments.

The objectives of the study are clear but the work would certainly attract attention from a broader audience by summarizing the main goal and specific objectives at the end of the introduction section.

RESPONSE: We have now extended the end of the Introduction to outline goals and objectives. See page 2.

It is clear that artificialized foods (notably class 4 of Nova classification) are unhealthy and unsustainable as shown by mounting scientific studies. True foods are dense in nutrients and bioactive compounds and its composition is complex and undetermined in opposition to such industrial foods, known as UPF. However, the definition of UPF has been contested, especially by engineers for whom “processing” is synonym of “unitary operations”. Other authors highlight the confusion with formulation. The authors are thus suggested to present these conflicting views (suggested ex. Levine AS, Ubbink J. Ultra-processed foods: Processing versus formulation. Obes Sci Pract. 2023 Jan 26;9(4):435-439. doi: 10.1002/osp4.657. Knorr D, Watzke H. Food Processing at a Crossroad. Front Nutr. 2019 Jun 25;6:85. doi: 10.3389/fnut.2019.00085.)

RESPONSE: We now make reference to some of the discussion surrounding NOVA and UPF, in addition to citing the suggested Levine reference. See page 1.

Please avoid less clear information and update the concept/definition supported by recent literature. Therefore, at this level, authors may consider deleting the examples in line 35 “Common examples of UPFs include confectionary, ice cream, breakfast cereals, packaged bread and reconstituted meat products (e.g., hot dogs).” Instead, “junk food” and “sodas” are more clearer examples to be presented at this level, if authors believe it to be useful.

Further clarification on class IV foods should be included at some level in the manuscript, because ice cream can be made of macerated fresh fruit, sacharose and often egg white and some corn flakes and oatmeal are processed in some way that are not included in class 4.

RESPONSE: The common examples we chose were from recent NOVA documentation for foods falling under the different classifications (Monteiro and colleagues). To address the reviewer comments we have removed reference in the introduction text to ice cream and breakfast cereal. We have added in sodas. When referring to the examples in the Introduction, we also make it clear what the source of the examples are (Monteiro and colleagues). See page 1.  

Lines 36-40 can also bring confusion as recently industries bring ultraprocessed ingredients to kitchens (ex. refined flour containing so-called “flour improvements” for home bread-making without the need for yeast fermentation). It can be also helpful for the reader if the authors also elaborate on processed foods, as those of class I, II, on nutritional security, convenience of use, extended shelflife vs. the poor nutritional features and negative impacts in public health and the environment of UPF.

RESPONSE: We have retained the text in this section because when defining UPF the most common classification system is NOVA and NOVA does not differentiate between processing that ensures nutritional security, convenience of use, shelf life, nutritional quality. However, we agree that these differentiations are important to consider and have added to the Discussion to outline that it will be important to understand which types of UPFs participants report avoiding consuming (e.g., those with poor nutritional features or not). See page 8.

Methods: Please clearly state inclusion and exclusion criteria as well as the number of initial recruited participants and those who competed the study, and the initial wording can thus be summarised.

RESPONSE: In the revised manuscript inclusion and exclusion criteria are outlined on page 2. Participants recruited and then included for analysis purposes (completers with full data) is reported on page 4.

Please include a scheme or table outlining the experimental design. Also include a table (different from that of supplementary material) clearly listing the different foods presented to participants, the corresponding short description, and the Nova class they belong to.

RESPONSE: The foods listed to participants were not at the brand level (e.g., burger brand), but at the food level (i.e., ‘burgers’). We have made this clearer in the manuscript text. Therefore, the text provided in the supplementary table provides the description participants saw, as well as the NOVA category for each food. We have edited the table for clarity and moved it to the main manuscript (Table 3). We have not included an additional table outlining the experimental design, as we did not feel the information added provided more detail than is covered in the manuscript text and we already have 4 tables. We have however elaborated on the experimental design on page 2.

Results: the discussion in the section UPF categorization can be enriched noticing the difficulties of participants in clearly understand the UPF concept and hence failing identifying class IV foods. Please elaborate and further explain results on table 3. Please also expand section 3.3. relating with the information from introduction (including newly inserted one). Please make the necessary adjustments on section 4.1 for coherence reasons. concluding remarks should be added; pls add a short conclusion section summarizing main findings, the relevance of the study and prospects.

RESPONSE: We have now added discussion on results relating to categorisation of UPFs (see end of page 7). We have further elaborated on findings from table 3 (now table 4). See page 8. Section 4.1 (limitations) has been adjusted in line with other reviewer comments. Results relating to section 3.3 are now outlined on page 8 of the discussion. We have also added in a final section entitled ‘conclusions’ and summarised the main findings there.
